# Association between Facial Emotion Recognition and Bullying Involvement among Adolescents with High-Functioning Autism Spectrum Disorder

**DOI:** 10.3390/ijerph16245125

**Published:** 2019-12-15

**Authors:** Tai-Ling Liu, Peng-Wei Wang, Yi-Hsin Connie Yang, Gary Chon-Wen Shyi, Cheng-Fang Yen

**Affiliations:** 1Department of Psychiatry, Kaohsiung Medical University Hospital, Kaohsiung 807, Taiwan; dai32155@gmail.com (T.-L.L.); wistar.huang@gmail.com (P.-W.W.); 2Department of Psychiatry, School of Medicine and Graduate Institute of Medicine, College of Medicine, Kaohsiung Medical University, Kaohsiung 807, Taiwan; 3Department of Pharmacy, College of Pharmacy, Kaohsiung Medical University, Kaohsiung 807, Taiwan; yhyang@nhri.edu.tw; 4National Institute of Cancer Research, National Health Research Institutes, Taipei 115, Taiwan; 5Department of Psychology and Center for Research in Cognitive Sciences, National Chung-Cheng University, Chiayi 621, Taiwan

**Keywords:** autism spectrum disorder, facial emotion recognition, bullying, adolescents

## Abstract

Autism spectrum disorder (ASD) is a neurodevelopmental disorder that is characterized by impaired social interaction, communication and restricted and repetitive behavior. Few studies have focused on the effect of facial emotion recognition on bullying involvement among individuals with ASD. The aim of this study was to examine the association between facial emotion recognition and different types of bullying involvement in adolescents with high-functioning ASD. We recruited 138 adolescents aged 11 to 18 years with high-functioning ASD. The adolescents’ experiences of bullying involvement were measured using the Chinese version of the School Bullying Experience Questionnaire. Their facial emotion recognition was measured using the Facial Emotion Recognition Task (which measures six emotional expressions and four degrees of emotional intensity). Logistic regression analysis was used to examine the association between facial emotion recognition and different types of bullying involvement. After controlling for the effects of age, gender, depression, anxiety, inattention, hyperactivity/impulsivity and opposition, we observed that bullying perpetrators performed significantly better on rating the intensity of emotion in the Facial Emotion Recognition Task; bullying victims performed significantly worse on ranking the intensity of facial emotion. The results of this study support the different deficits of facial emotion recognition in various types of bullying involvement among adolescents with high-functioning ASD. The different directions of association between bully involvement and facial emotion recognition must be considered when developing prevention and intervention programs.

## 1. Introduction

School bullying is a critical problem concerning the psychological health of adolescents. Bullying behaviors include physical attacks (e.g., hitting, pushing, or kicking), verbal attacks (e.g., calling names, spreading rumors, or threatening somebody), and intentional actions that cause the victim to experience social isolation [1]. Different types of involvement in bullying produce different risks of mental health problems among adolescents [2]. Adolescents who are victims of school bullying exhibit poorer psychological health [3], higher levels of depression [3] and anxiety [4], and higher risks of suicide [5] than general adolescents do. Olweus [6] also reported that people who were constantly bullied during their teenage years demonstrate more severe problems regarding low self-esteem and depression once they reach adulthood. Although Volk et al. proposed that bullying perpetration may be the result of an evolutionarily adaption for somatic resources, mates, and dominance [7], some researchers argued that both pure bullying perpetrators and perpetrator-victims have higher risks to develop mental health problems [2,8]. A previous study also reported that both pure perpetrators and perpetrator-victims were more likely to report suicidal ideation and attempt than the neutral group [9]. Therefore, school bullying is an important topic and requires early intervention.

Autism spectrum disorder (ASD) is a neurodevelopmental disorder associated with impediments in social interaction and communication and a tendency to demonstrate restricted interests and repetitive behaviors [10]. Various studies have indicated that adolescents with ASD are more likely to be involved in bullying compared with those without ASD [11,12]. A review article by Maïano et al. [13] stated that school-aged youth with ASD are at greater risks of school victimization and verbal bullying than their peers without ASD.

Research has proposed possible etiologies to explain why adolescents with ASD are more likely to be involved in bullying, including having fewer friends [14], tending to demonstrate repetitive and stereotypical behaviors [15], having comorbid intellectual disability [16], and being more likely to engage in aggressive behaviors [17]. Overall, bullying involvement may result from or in social interaction impairment in individuals with ASD.

Several studies have indicated that the social interaction impairment of individuals with ASD might be partially caused by their inability to recognize facial emotions [18,19,20,21,22]. Facial emotion recognition refers to the ability to correctly identify and express nonverbal social cues and serves as a crucial factor of favorable social interactions [19]. Impaired ability to recognize facial emotions can lead to difficulty engaging in social interactions and can even encourage aggressive behaviors. Studies have found that compared with individuals without ASD, those with high-functioning ASD demonstrate significantly poorer ability to recognize negative facial emotions such as sadness [20] and fear [21], in addition to complex facial expressions such as guilt, embarrassment, and jealousy [22]. However, no study has examined the correlation between facial emotion recognition and involvement in bullying among individuals with ASD. In addition, whether impairment in facial emotion recognition varies among ASD adolescent bullying perpetrators, victims, and neutrals should be determined. If facial emotion recognition could be verified as a risk factor for bullying involvement in adolescents with ASD, prevention and intervention strategies for bullying involvement should include programs for enhancing the ability of facial emotion recognition.

Dyck et al. [23] reported that the age, communication ability, and reception language of children with ASD could predict their ability in facial emotion recognition. Moreover, students with high-functioning ASD typically study in general classes because they are of normal intelligence. For such students, the types of bullying involvement may be different from those encountered by students with ASD comorbid intellectual disability [24]. Therefore, the present study focused on the experiences of bullying involvement in adolescents with high-functioning ASD. 

The aim of this study was to examine the correlation between facial emotion recognition and bullying involvement in adolescents with high-functioning ASD. Previous studies regarding teenagers’ bullying have indicated that bullying roles and behaviors are diverse. Different bullying behaviors involve teenagers’ skills, status, and social behaviors [25]. Sutton et al. [26] claimed that a portion of the bullying perpetrators are of high social intelligence and have superior theory of mind, because lack of social intelligence often results in ineffective bullying behaviors [26]. Peeters et al. [25] also identified that a portion of the bullying perpetrators had higher social cognition than the victims did. Therefore, the perpetrators and victims may exhibit different social cognition levels [25]. This study divided bullying roles into perpetrators and victims for further exploration. We hypothesized that adolescents with high-functioning ASD who were bullying perpetrators had better facial emotion recognition than those who were non-perpetrators, whereas high-functioning ASD who were bullying victims had worse facial emotion recognition than those who were non-victims.

ASD is often comorbid with attention-deficit hyperactivity disorder (ADHD), depression, and anxiety. Previous studies have discovered that ADHD [2,27], depression [2,28], and anxiety [2,29] may be risk factors of bullying involvement and facial emotion misrecognition. In addition, sociodemographic data, such as sex [30] and age [31,32], are often considered relevant to bullying involvement. Therefore, this study controlled for sex, age, and severity of ADHD, depression, and anxiety to identify the correlation between facial emotion recognition and bullying involvement.

## 2. Methods

### 2.1. Participants

A total of 210 adolescents with ASD visited the child psychiatry outpatient clinic of a university hospital in southern Taiwan during the period between October 2015 and July 2017. Adolescents with high-functioning ASD from were enrolled from this group for this study. The inclusion criteria were as follows: (1) having an age of 11–18 years; (2) having a diagnosis of ASD ascertained by a certified child psychiatrist according to the Diagnostic and Statistical Manual of Mental Disorders, Fifth Edition (DSM-5; [10]); (3) having a full-scale intelligence quotient score of 70 or higher, determined using the Chinese version of the Wechsler Intelligence Scale for Children, Fourth Edition (WISC-IV; [33]); and (4) having verbal communication ability. Individuals who had severe comorbid physical problems, had a history of severe brain injury or substance abuse, were comorbid with other severe psychiatric disorders, or were uncooperative to complete all evaluations were excluded. A total of 142 adolescents with high-functioning ASD were initially invited to this study—however, four did not complete the tests. Finally, the data of 138 participants (124 boys and 14 girls) were used for analysis. Their average age was 13.87 years (standard deviation (SD): 1.51 years).

### 2.2. Measures

#### 2.2.1. The Computerized Facial Emotion Recognition Task (C-FERT)

The development of the C-FERT has been described elsewhere [34,35]. In brief, the C-FERT comprises 70 pictures of Taiwanese people (38 women and 32 men) depicting six categories of basic emotion (happiness, sadness, disgust, fear, anger, and surprise) and neutral emotion [34,35]. Intensities of emotion depicted in the pictures were classified as mild (30% intensity), moderate (60% intensity), and strong (90% intensity). The C-FERT was administered in three parts. In the first part (C-FERT differentiation), participants were shown one randomly selected picture from 70 pictures and were then asked to push a button as soon as possible to select the most appropriate emotion displayed. The picture was shown for a maximum of 5 s. The computer recorded the reaction times and correct rates of 20 pictures with different intensities randomly selected. In the second part (C-FERT ranking), three pictures showing the same category of emotion with different intensities (mild, moderate, and strong) were displayed simultaneously on the screen. Participants were asked to sequence the pictures in order of emotional intensity by selecting mild, moderate, or strong intensity within 10 s. Ten groups of pictures were shown, and each correct rate was recorded. In the third part (C-FERT rating), 10 pictures were selected randomly by the computer, and participants were requested to rate the emotional intensity represented in the pictures from 0 (“neutral”) to 3 (“strongest intensity”), with the computer recording reaction times and correct rates. The total test duration of the C-FERT lasted approximately 20–30 min.

#### 2.2.2. Chinese Version of School Bullying Experience Questionnaire (C-SBEQ)

The C-SBEQ contains 16 items; the first eight items concern experiences of being bullied, and the final eight items concern the bullying of others in school within the preceding year. A 4-point Likert scale was adopted for all items, and participants were required to choose from the following four options according to the severity level (0: “never”; 1: “occasionally”; 2: “often”; and 3: “always”). This scale was composed of four 4 item subscales for evaluating the severity of victimization by verbal and relational bullying (Items 1–4, including social exclusion, being called mean nicknames, and being spoken ill of, e.g., “Are you left out during recess or lunch time?”), victimization by physical bullying and snatching of belongings (Items 5–8, including being beaten up, being forced to do others’ work, and having money, school supplies, or snacks taken away, e.g., “Are you beaten up?”), perpetration of verbal and relational bullying (Items 9–12, e.g., “Do you leave out other students during recess or lunch time?”), perpetration of physical bullying and snatching of belongings (Items 13–16, e.g., “Do you beat up other students?”). Participants who answered 2 or 3 on any item among items 1 to 8 and items 9 to 16 were identified as bullying victims and perpetrators, respectively. The C-SBEQ has excellent reliability and validity [36,37]. In the present study the Cronbach’s α coefficient ranged from 0.68 to 0.74 for the four subscales.

#### 2.2.3. Chinese Version of the Center for Epidemiological Studies-Depression Scale (C-CES-D)

The C-CES-D consists of a self-report questionnaire with 20 items investigating symptoms related to depression such as emotions, appetite, sleep, sadness, loneliness, worthlessness (e.g., “I thought my life had been a failure.”), fatigue, and social withdrawal (e.g., “I felt that people dislike me.”). The C-CES-D was edited by Chien and Cheng (1985) and is outstanding in sensitivity and specificity [38]. Total scores ranged from 0 to 60, and higher C-CES-D total scores indicated more severe depression [38]. The Cronbach’s alpha for the C-CES-D in the present study was 0.93.

#### 2.2.4. Taiwanese Version of the Multidimensional Anxiety Scale for Children (MASC-T)

The MASC-T consists of 39 items measuring multidimensional anxiety symptoms, including physical symptoms (tense/restless and somatic/autonomic), harm avoidance (perfectionism and anxious coping), social anxiety (humiliation/rejection and public performance fears) and separation anxiety, and panic in children and adolescents aged 8 to 19 years [39,40]. A higher total score of MASC-T demonstrates a higher anxiety level. The Cronbach’s alpha for the MASC-T in the present study was 0.89.

#### 2.2.5. Swanson, Nolan, and Pelham, Version IV Rating Scale (SNAP-IV)

The 26 item SNAP-IV assesses the severity of inattention (e.g., “Often is distracted by extraneous stimuli”), hyperactivity/impulsivity (e.g., “Often talks excessively”), and oppositional defiant symptoms (e.g., “Often argues with adults”) [41]. The parent-reported version of SNAP-IV in Chinese has outstanding reliability and validity and has been recognized as a standard in Taiwan. The Chinese SNAP-IV demonstrated satisfactory internal consistency, concurrent validity, and discriminant validity [42]. The Cronbach’s alpha for the three dimensions of the Chinese SNAP-IV in the present study ranged from 0.75 to 0.88.

### 2.3. Procedure and Statistical Analysis

Participants with ASD referred by outpatient clinics were individually provided with an explanation of the purpose, procedure, and privacy policy of this study by the research assistant before the participants signed a consent form. The participants’ parents were asked to respond to the questionnaire for sociodemographic data and SNAP-IV; adolescents were asked to respond to the C-SBEQ, C-CES-D, MASC-T, and C-FERT. The overall response process required approximately 35–45 min. Data analysis was performed using SPSS 22.0 statistical software (SPSS Inc., Chicago, IL, USA).

In the initial exploration of the data we first divided the participants into four groups, namely pure perpetrators, pure victims, perpetrators/victims, and neutrals (Table 1). The results revealed that the differences in C-FERT among the four groups did not reach the significant level. It was possibly caused by the small numbers of participants in the groups of perpetrators/victims and pure victims. Therefore, we regrouped the participants and analyzed facial emotion recognition between bullying perpetrators and non-perpetrators as well as between bullying victims and non-victims. Participants who answered 2 or 3 on any item among items 1 to 8 and items 9 to 16 from C-SBEQ were identified as bullying victims and perpetrators, respectively.

Finally, all participants were divided into the perpetrator group or victim group and the nonperpetrator group or nonvictim group to further analyze the differences between both groups. Differences in demographic data, depression, anxiety, inattention, hyperactivity/impulsivity, opposition, and facial emotion recognition between bullying perpetrators and nonperpetrators and between bullying victims and nonvictims were examined using a chi-square test and *t*-test. Logistic regression analysis was used to examine the associations between facial emotion recognition and being bullying perpetrators and victims by controlling for the effects of other factors. A two-tailed *p* value of <0.05 was considered statistically significant.

## 3. Results

In total, 138 adolescents with high-functioning ASD completed this study. Among them, 41 (29.71%) were classified as bullying perpetrators and 24 (17.39%) were classified as victims. Table 2 presents the results obtained from comparing demographic data, depression, anxiety, inattention, hyperactivity/impulsivity, opposition, and facial emotion recognition between bullying perpetrators and nonperpetrators. The results revealed that compared with nonperpetrators, bullying perpetrators had more severe depression and anxiety. The two groups exhibited significant differences in C-FERT rating performance. Bullying perpetrators were notably superior to the nonperpetrators regarding the correct rate and reaction time in rating the intensity of facial emotion, indicating that bullying perpetrators had the superior ability to recognize facial expressions compared with nonperpetrators.

Table 3 shows the results obtained from comparing demographic data, depression, anxiety, inattention, hyperactivity/impulsivity, opposition, and facial emotion recognition between bullying victims and nonvictims. The results showed that compared with nonvictims, bullying victims exhibited higher levels of depression and anxiety. Additionally, the two groups did not exhibit significant differences in facial emotion recognition.

Table 4 and Table 5 present the results of logistic regression analysis examining the association of facial emotion recognition with being bullying perpetrators and victims, respectively. In Model I to Model V, the correct rate and reaction time of C-FERT differentiation, correct rate of C-FERT ranking, as well as correct rate and reaction time of C-FERT rating were selected to multiple logistic regression as independent variables, respectively. Age, gender, depression, anxiety, inattention, hyperactivity/impulsivity and opposition were selected as control variables. The results of Models IV and V in Table 4 indicated that bullying perpetrators performed significantly better on rating intensity of facial emotion in the C-FERT rating process than nonperpetrators did. The results of Model III in Table 5 indicated that bullying victims performed significantly worse on ranking intensity of facial emotion in the C-FERT ranking process than nonvictims.

## 4. Discussion

This study is the first to examine the association between the ability of facial emotion recognition and bullying perpetration and victimization in adolescents with high-functioning ASD. The study found that bullying perpetrators with high-functioning ASD had a greater ability to correctly rate the intensity of facial emotion than nonperpetrators did. Because perpetrators with high-functioning ASD may have a greater ability to recognize the expressions of others, they recognized how to target weaker people and successfully bully them. By contrast, victims with high-functioning ASD had a lower ability to rank the intensity of facial emotion than nonvictims did. Poor ability to recognize the intensity of others’ facial emotions may make adolescents with ASD less sensitive to others’ emotional changes and increase their possibility to be considered impolite, thus subsequently increasing their risk of being bullied.

Previous studies investigating the relationship between bullying involvement and the ability to recognize facial emotion have focused primarily on adolescents with typical neurodevelopment rather than adolescents with ASD. Most of these studies on general children and adolescents have discovered that victims had a lower ability to correctly recognize others’ facial emotions [43,44]; however, research has found no significant difference in emotion recognition abilities between bullying perpetrators and neutrals [45,46]. Pozzoli et al. [44] studied adolescents with typical neurodevelopment and observed that bullying perpetrators were more likely to correctly recognize others’ emotion of fear than nonperpetrators did, allowing them to more efficaciously perform aggressive behaviors. By contrast, bullying victims were less able to correctly recognize others’ emotions of anger and disgust, which may increase their risk of being targeted and bullied [44]. The present study on adolescents with high-functioning ASD has similar findings to those of Pozzoli et al. [44] and provides a preliminary understanding of the relationship between the ability of facial emotion recognition and bullying involvement in adolescents with high-functioning ASD. Further study is required to investigate the mechanisms that account for the significant relationship.

Previous interventions against school bullying have often used universal bullying prevention programs (e.g., the Olweus Bullying Prevention Program) as the criterion standard [47]. Other recent intervention modes, including the “Learning Together” intervention program posed by Bonell et al. [48], have focused on improving social and emotional skills. However, these programs did not anchor with the core symptoms of ASD. Research found that the training program focusing on enhancing theory of mind performance ability could reduce the risk of bullying victimization in children and adolescents with high-functioning ASD [49]. Facial emotion recognition is one of the skills trained to enhance theory of mind performance ability [49]. The present study found that poor ability to recognize facial emotion was related to bullying victimization in adolescents with high-functioning ASD. Therefore, researchers should consider adding facial emotion recognition training to prevention and intervention programs for reducing the risk of bullying victimization in adolescents with high-functioning ASD. Moreover, based on the results of this study, the possibility of perpetrating bullying should be monitored in adolescents with high-functioning ASD who have a better ability to recognize facial emotion.

The present study adjusted for the effects of age, gender, depression, anxiety, inattention, hyperactivity/impulsivity and opposition when examining the relationship between the ability of facial emotion recognition and bullying involvement in adolescents with high-functioning ASD. The study also used the facial emotion pictures of Taiwanese rather than Western people to reduce ASD participants’ unfamiliarity. Despite these adjustments, this study has some limitations that should be addressed. First, the cross-sectional research design of this study limited our ability to draw conclusions regarding the causal relationships between facial emotion recognition and bullying involvement. Second, this study evaluated bullying involvement based on adolescents’ self-reports. Future studies are required to identify whether the results differ based on bullying involvement reported by adolescents’ parents and teachers. Third, this study recruited the participants from outpatient clinics; therefore, many exhibited emotional or behavioral disturbances. Determining whether the results of the present study can be generalized to all adolescents with ASD requires further study. Fourth, the C-FERT used in this study was a random test of pictures of facial expressions. In addition, whether difficulties in specific types of facial emotion recognition exist in relation to bullying perpetration and victimization warrants further examination. Moreover, the small number of participants who were both bullying perpetrators and victims (perpetrator-victims) limited the possibility of this study to examine whether the perpetrator-victims possess unique patterns of facial emotion recognition. Further study should be conducted to realize the role of facial emotion recognition ability in relation to bullying involvement among adolescents with high-functioning ASD when bullying involvement is divided into four categories.

## 5. Conclusions

This study on adolescents with high-functioning ASD revealed that bullying perpetrators possessed a better ability to recognize facial emotion than nonperpetrators did, whereas bullying victims had worse ability to recognize facial emotion than nonvictims did. The ability to recognize facial emotion should be considered when developing prevention and intervention programs for bullying involvement in adolescents with ASD.

## Figures and Tables

**Table 1 ijerph-16-05125-t001:** Comparisons of demographic data, depression, anxiety, inattention, hyperactivity/impulsivity, opposition, and facial emotion recognition among the four groups with various experiences of bullying involvement.

	Perpetrators (*n* = 29)		Victims (*n* = 12)			Perpetrators/Victims (*n* = 12)		Neutrals (*n* = 85)		
	*n* (%)	Mean (SD)	*n* (%)	Mean (SD)	*n* (%)	Mean (SD)	*n* (%)	Mean (SD)	F	*p*
Gender									0.493	0.688
Male	27 (93)		10 (83)		10 (83)		77 (91)			
Female	2 (7)		2 (17)		2 (17)		8 (9)			
Age (years)		13.96 (2.10)		14.35 (1.93)		14.04 (2.52)		13.75 (2.20)	0.321	0.810
Father’s education duration (years)		14.22 (3.07)		14.91 (2.47)		13.50 (3.42)		14.65 (2.88)	0.686	0.562
Mother’s education duration (years)		12.79 (1.89)		14.17 (2.21)		13.64 (2.16)		13.80 (2.87)	1.268	0.288
Depression		18.28 (9.96)		17.92 (4.85)		24.92 (9.35)		13.36 (6.34)	10.498	<0.001
Anxiety		52.31 (10.25)		54.25 (9.35)		59.67 (12.93)		46.25 (10.32)	8.022	<0.001
Inattention		75.31 (28.99)		87.25 (14.78)		81.42 (23.66)		80.88 (21.36)	0.854	0.467
Hyperactivity/impulsivity		67.83 (27.38)		66.00 (34.52)		75.17 (26.33)		66.75 (26.61)	0.344	0.793
Opposition		75.41 (25.98)		79.25 (28.55)		75.00 (28.69)		78.14 (21.10)	0.163	0.921
C-FERT differentiation, correct rate (%)		76.53 (9.84)		74.11 (16.42)		76.12 (10.81)		73.73 (12.90)	0.433	0.730
C-FERT differentiation, reaction time (s)		2.20 (0.88)		2.31 (6.14)		2.24 (0.38)		2.31 (0.63)	0.198	0.898
C-FERT ranking, correct rate (%)		80.02 (17.64)		68.83 (25.09)		71.98 (18.94)		78.52 (16.41)	1.629	0.186
C-FERT rating, correct rate (%)		49.34 (16.18)		43.94 (17.26)		55.68 (8.70)		44.24 (14.68)	2.648	0.052
C-FERT rating, reaction time (s)		1.71 (0.34)		1.95 (0.45)		1.85 (0.40)		1.92 (0.51)	1.685	0.173

C-FERT: Computerized Facial Emotion Recognition Test.

**Table 2 ijerph-16-05125-t002:** Comparisons of demographic data, depression, anxiety, inattention, hyperactivity/impulsivity, opposition, and facial emotion recognition between bullying perpetrators and non-perpetrators.

	Perpetrators (*n* = 41)		Non-Perpetrators (*n* = 97)				
	*n* (%)	Mean (SD)	*n* (%)	Mean (SD)	χ^2^ or *t*	*p*	Cohen’s d
Gender							
Male	37 (90)		87 (93)		0.010	0.922	
Female	4 (10)		10 (7)				
Age (years)		13.98 (2.20)		13.82 (2.16)	0.395	0.694	
Father’s education duration (years)		14.00 (3.15)		14.69 (2.82)	−1.225	0.223	
Mother’s education duration (years)		13.03 (1.98)		13.84 (2.79)	−1.911	0.059	
Depression		20.22 (10.14)		13.93 (6.34)	3.681	**0.001**	0.744
Anxiety		54.46 (11.45)		47.24 (10.50)	3.597	**<0.001**	0.657
Inattention		77.10 (27.39)		81.61 (20.70)	−0.959	0.341	
Hyperactivity/impulsivity		69.98 (26.96)		66.66 (27.50)	0.651	0.516	
Opposition		75.29 (26.44)		78.28 (21.98)	−0.686	0.494	
C-FERT differentiation, correct rate (%)		76.41 (10.00)		73.78 (13.29)	1.140	0.256	
C-FERT differentiation, reaction time (s)		2.22 (0.76)		2.31 (0.62)	−0.752	0.454	
C-FERT ranking, correct rate (%)		77.66 (18.17)		77.32 (17.84)	0.102	0.919	
C-FERT rating, correct rate (%)		51.20 (14.85)		44.21 (14.92)	2.532	**0.012**	0.470
C-FERT rating, reaction time (s)		1.75 (0.36)		1.93 (0.50)	−2.369	**0.020**	0.413

C-FERT: Computerized Facial Emotion Recognition Test.

**Table 3 ijerph-16-05125-t003:** Comparisons of demographic data, depression, anxiety, inattention, hyperactivity/impulsivity, opposition, and facial emotion recognition between bullying victims and non-victims.

	Victims (*n* = 24)		Non-Victims (*n* = 114)				
	*n* (%)	Mean (SD)	*n* (%)	Mean (SD)	χ^2^ or *t*	*p*	d
Gender							
Male	20 (83)		104 (91)		1.356	0.244	
Female	4 (17)		10 (9)				
Age (years)		14.20 (2.20)		13.80 (2.17)	0.807	0.421	
Father’s education duration (years)		14.18 (3.02)		14.55 (2.92)	−0.552	0.582	
Mother’s education duration (years)		13.91 (2.15)		13.54 (2.69)	0.624	0.534	
Depression		21.42 (8.11)		14.61 (7.69)	3.903	**<0.001**	0.862
Anxiety		56.96 (11.38)		47.79 (10.59)	3.805	**<0.001**	0.834
Inattention		84.33(19.52)		79.46(23.52)	0.947	0.345	
Hyperactivity/impulsivity		70.58(30.39)		67.03(26.69)	0.579	0.564	
Opposition		77.13 (28.08)		77.45 (22.35)	−0.061	0.951	
C-FERT differentiation, correct rate (%)		75.12 (13.63)		74.44 (12.22)	0.240	0.810	
C-FERT differentiation, reaction time (s)		2.28 (0.50)		2.28 (0.70)	−0.024	0.981	
C-FERT ranking, correct rate (%)		70.40 (21.80)		78.90 (16.67)	−1.802	0.082	
C-FERT rating, correct rate (%)		49.81 (14.65)		45.54 (15.17)	1.261	0.209	
C-FERT rating, reaction time (s)		1.90 (0.42)		1.87 (0.48)	0.247	0.805	

C-FERT: Computerized Facial Emotion Recognition Test.

**Table 4 ijerph-16-05125-t004:** The association of facial emotion recognition with being bullying perpetrators after controlling age, gender, depression and anxiety, inattention, hyperactivity/impulsivity and opposition.

	Model I		Model II		Model III		Model IV		Model V	
	OR (95%)	*p*	OR (95%)	*p*	OR (95%)	*p*	OR (95%)	*p*	OR (95%)	*p*
Gender	1.721 (0.382–7.756)	0.480	1.969 (0.433–8.739)	0.373	1.781 (0.401–7.903)	0.448	2.302 (0.441–12.071)	0.323	1.302 (0.269–6.300)	0.743
Age	0.972 (0.796–1.187)	0.780	0.970 (0.792–1.190)	0.773	.964 (0.790–7.903)	0.717	0.992 (0.907–1.218)	0.936	0.970 (0.786–1.197)	0.774
Depression	1.096 (1.021–1.175)	0.011	1.100 (1.024–1.181)	0.009	1.097 (1.022–1.176)	0.010	1.101 (1.025–1.182)	0.008	1.121 (1.038–1.210)	0.003
Anxiety	1.032 (0.979–1.087)	0.244	1.035 (0.983–1.091)	0.193	1.032 (0.980–1.087)	0.227	1.031 (.976–1.090)	0.277	1.031 (0.978–1.088)	0.257
Inattention	0.978 (0.955–1.001)	0.057	0.977 (0.955–1.001)	0.056	0.977 (0.955–1.000)	0.053	0.974 (0.951–0.998)	0.034	0.981 (0.958–1.005)	0.121
Hyperactivity/impulsivity	1.024 (1.000–1.049)	0.051	1.026 (1.001–1.051)	0.043	1.025 (1.000–1.050)	0.050	1.022 (0.996–1.049)	0.092	1.019 (0.994–1.044)	0.131
Opposition	0.980 (0.957–1.003)	0.084	0.980 (0.957–1.003)	0.091	0.979 (0.956–1.002)	0.074	0.983 (0.960–1.007)	0.175	0.984 (0.961–1.008)	0.198
C-FERT differentiation, correct rate	1.013 (0.977–1.050)	0.477								
C-FERT differentiation, reaction time			1.000 (0.999–1.000)	0.181						
C-FERT ranking, correct rate					1.004 (0.981–1.028)	0.737				
C-FERT rating, correct rate							1.038 (1.005–1.072)	**0.024**		
C-FERT rating, reaction time									0.999 (0.997–1.000)	**0.011**
−2 log likelihood	140.417		139.030		140.822		135.044		133.699	
Nagelkerke R^2^	0.257		0.268		0.253		0.301		0.312	
Wald χ^2^	27.498		28.885		27.093		32.871		34.216	

C-FERT: Computerized Facial Emotion Recognition Test; OR: odds ratio.

**Table 5 ijerph-16-05125-t005:** The association of facial emotion recognition with being bullying victims after controlling age, gender, depression and anxiety, inattention, hyperactivity/impulsivity and opposition.

	Model I		Model II		Model III		Model IV		Model V	
	OR (95%)	*p*	OR (95%)	*p*	OR (95%)	*P*	OR (95%)	*p*	OR (95%)	*p*
Gender	0.610 (0.147–2.532)	0.496	0.639 (0.157–2.595)	0.531	0.693 (0.169–2.837)	0.610	0.649 (0.153–2.760)	0.559	0.571 (0.135–2.411)	0.445
Age	0.958 (0.756–1.213)	0.720	0.957 (0.755–1.214)	0.718	0.958 (0.750–1.226)	0.735	0.960 (0.758–1.217)	0.739	0.953 (0.751–1.210)	0.694
Depression	1.056 (0.983–1.134)	0.134	1.057 (0.984–1.135)	0.127	1.049 (0.976–1.128)	0.193	1.058 (0.985–1.137)	0.121	1.060 (0.986–1.140)	0.116
Anxiety	1.050 (0.982–1.111)	0.092	1.052 (0.933–1.113)	0.083	1.059 (1.000–1.121)	0.052	1.048 (0.989–1.110)	0.116	1.050 (0.992–1.111)	0.092
Inattention	1.014 (0.984–1.045)	0.366	1.014 (0.984–1.045)	0.358	1.013 (0.982–1.044)	0.417	1.013 (0.984–1.044)	0.380	1.015 (0.985–1.046)	0.331
Hyperactivity/impulsivity	1.005 (0.980–1.029)	0.708	1.006 (0.981–1.031)	0.659	1.003 (0.979–1.028)	0.826	1.003 (0.979–1.029)	0.794	1.003 (0.978–1.028)	0.816
Opposition	0.983 (0.958–1.008)	0.179	0.983 (0.958–1.008)	0.184	0.987 (0.962–1.013)	0.326	0.984 (0.959–1.010)	0.226	0.984 (0.959–1.010)	0.216
C-FERT differentiation, correct rate	1.003 (0.963–1.044)	0.880								
C-FERT differentiation, reaction time			1.000 (0.999–1.001)	0.560						
C-FERT ranking, correct rate					0.976 (0.952–1.000)	**0.048**				
C-FERT rating, correct rate							1.014 (0.982–1.047)	0.402		
C-FERT rating, reaction time									1.000 (0.999–1.001)	0.567
−2 log likelihood	109.816		109.487		105.999		109.108		109.506	
Nagelkerke R^2^	0.200		0.203		0.239		0.207		0.203	
Wald χ^2^	17.706		18.035		21.523		18.414		18.016	

C-FERT: Computerized Facial Emotion Recognition Test; OR: odds ratio.

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
