# Peer review of "Association between Facial Emotion Recognition and Bullying Involvement among Adolescents with High-Functioning Autism Spectrum Disorder"

_ijerph, 2019, doi:10.3390/ijerph16245125_

Round 1
Reviewer 1 Report
The manuscript needs some major revision before it can be publish.
1) Given that the analysis are separately conducted for victims and perpetrators of victims, hypothesis should be formulated for each role. They should be sure to state these in ways that do allow direct testing and refer to the actual variables being used (now it is not explained what exactly expected the authors. For example, they stated that deficits in facil emotion recognition might be different between victims and perpetrators. How different? In what direction?). They should make sure that any variables that appear in the later analyses are included in at least one of these hypotheses. Authors must also explain in what grounds are based each hypothesis.
2) Participants description should be improved. For example, what was the whole sample of adolescents with ASD before the exclusion attending to the study criteria? How many individuals were excluded from the study?. Mean age of the participants? gender distribution?
3) Instruments: item examples should be included in each measure. Reliability in the current sample for each scale should be included.
4) Categorization procedures of bullying victims and perpetrators should be explained again in the statistical analysis section.
5) Statistics on the effect size should be added in table 1 and 2 in those variables were differences were statistical significant.
6) Authors should include statistical data like (-2 LL), Nagelkerke R2, Value of the Model X² in the logistic regressions in order to understand the significance of the results.
7) In the abstract, authors stated that the different directions of association between bully involvement and facial emotion recognition must be considerd when developing prevention and intervention programs. However, they did not address this point in the discussion section. How facial emotion recognition should be considered? Are there already programs working with facial emotion recognition in samples of ASD and adolescent with typical neurodevelopment? Given that the journal is focus on public health, what are the implications of the study in those terms?
Author Response
Comment
Given that the analysis are separately conducted for victims and perpetrators of victims, hypothesis should be formulated for each role. They should be sure to state these in ways that do allow direct testing and refer to the actual variables being used (now it is not explained what exactly expected the authors. For example, they stated that deficits in facial emotion recognition might be different between victims and perpetrators. How different? In what direction?). They should make sure that any variables that appear in the later analyses are included in at least one of these hypotheses. Authors must also explain in what grounds are based each hypothesis.Response
Thank you for your suggestion. We have revised the introduction section to add to reasons of hypotheses and describe the hypotheses with the hypothesized directions as below. Please refer to line 85-97.“Previous studies regarding teenagers’ bullying have indicated that bullying roles and behaviors are diverse. Different bullying behaviors involve teenagers’ skills, status, and social behaviors [23]. Sutton et al. [24] claimed that a portion of the bullying perpetrators are of high social intelligence and have superior theory of mind, because lack of social intelligence often results in ineffective bullying behaviors [24]. Peeters et al. [23] also identified that a portion of the bullying perpetrators had higher social cognition than the victims did. Therefore, the perpetrators and victims may exhibit different social cognition levels [23]. So this study divided bullying roles into perpetrators and victims for further exploration. We hypothesized that adolescents with high-functioning ASD who were bullying perpetrators had better facial emotion recognition than those who were non-perpetrators, whereas high-functioning ASD who were bullying victims had worse facial emotion recognition than those who were non-victims.”
We also added the reasons why we considered sex, age, and severity of ADHD, depression, and anxiety as covariates in the present study as below. Please refer to line 98-105.“ASD is often comorbid with attention deficit hyperactivity disorder (ADHD), depression, and anxiety. Previous studies have discovered that ADHD [2, 25], depression [2, 26], and anxiety [2, 27] may be risk factors of bullying involvement and facial emotion misrecognition. In addition, sociodemographic data, such as sex [28] and age [29, 30], are often considered relevant to bullying involvement. Therefore, this study controlled for sex, age, and severity of ADHD, depression, and anxiety to identify the correlation between facial emotion recognition and bullying involvement.”
Comment
Participants description should be improved. For example, what was the whole sample of adolescents with ASD before the exclusion attending to the study criteria? How many individuals were excluded from the study? Mean age of the participants? gender distribution?Response
Thank you for your suggestion. We have revised the methods section and introduced the whole sample of adolescents with ASD before the exclusion and the mean age and gender distribution of the participants as below.
“A total of 210 adolescents with ASD visited the child psychiatry outpatient clinic of a university hospital in southern Taiwan during the period between October 2015 and July 2017.” (Line 109-111) “A total of 142 adolescents with high-functioning ASD were invited into this study initially. Of them, four did not complete the tests. Finally, the data of 138 participants (124 boys and 14 girls) were used for analysis. Their average age was 13.87 years (standard deviation [SD]: 1.51 years).” (Line 121-124)
Comment
Instruments: item examples should be included in each measure. Reliability in the current sample for each scale should be included.Response
Thank you for your suggestion. We have revised the methods section to include item examples in each measure and reliability in the current sample for each scale as below.
C-SBEQ: “This scale was composed of four 4-item subscales for evaluating the severity of victimization by verbal and relational bullying (Items 1–4, including social exclusion, being called mean nicknames, and being spoken ill of, e.g., “Are you left out during recess or lunch time?”), victimization by physical bullying and snatching of belongings (Items 5–8, including being beaten up, being forced to do others’ work, and having money, school supplies, or snacks taken away, e.g., “Are you beaten up?”), perpetration of verbal and relational bullying (Items 9–12, e.g., “Do you leave out other students during recess or lunch time?”), perpetration of physical bullying and snatching of belongings (Items 13–16, e.g., “Do you beat up other students?”). (Line 153-163)…”In the present study the Cronbach’s α coefficient ranged from .68 to .74 for the four subscales.” (Line 165-167) C-CES-D: “(g., “I thought my life had been a failure.”)”…(e.g., “I felt that people dislike me.”)” (Line 172 and 173)….“Total scores ranged from 0 to 60, and higher C-CES-D total scores indicated more severe depression [36]. The Cronbach’s alpha for the C-CES-D in the present study was .93.” (Line 175-177) MASC-T: “(tense/restless and somatic/autonomic)”…”(perfectionism and anxious coping)”…”(humiliation/rejection and public performance fears)” (Line 180-182)…”The Cronbach's alpha for the MASC-T in the present study was .89.” (Line 184-185) Chinese SNAP-IV: “(g., “Often is distracted by extraneous stimuli”)…(e.g., “Often talks excessively”)…(e.g., “Often argues with adults”)” (Line 187-189)…“The Chinese SNAP-IV demonstrated satisfactory internal consistency, concurrent validity, and discriminant validity [40]. The Cronbach's alpha for the three dimensions of the Chinese SNAP-IV in the present study ranged from .75 to .88.” (Line 192-194)
Comment
Categorization procedures of bullying victims and perpetrators should be explained again in the statistical analysis section.Response
Thank you for your suggestion. We have added the explanation for the categorizing procedures as below in methods section Please refer to line 204-213. We also added a new table (Table 1) to explain why we adjusted the categories into two groups of perpetrators/non perpetrators and victims/non victims for further analysis.
“In the initial exploration of the data we first divided the participants into four groups, namely pure perpetrators, pure victims, perpetrators/victims, and neutrals (Table 1). The results revealed that the differences in C-FERT among the four groups did not reach the significant level. It was possibly caused by the small numbers of participants in the groups of perpetrators/victims and pure victims. Therefore, we regrouped the participants and analyzed facial emotion recognition between bullying perpetrators and non-perpetrators as well as between bullying victims and non-victims. Participants who answered 2 or 3 on any item among items 1 to 8 and items 9 to 16 from C-SBEQ were identified as bullying victims and perpetrators, respectively.”
Comment
Statistics on the effect size should be added in table 1 and 2 in those variables were differences were statistically significant.Response
Thank you for your suggestion. We have added Cohen’s d into Tables 2 and 3 (original Tables 1 and 2).
Comment
Authors should include statistical data like (-2 LL), Nagelkerke R2, Value of the Model X² in the logistic regressions in order to understand the significance of the results.Response
Thank you for your suggestion. We have added -2LL, Nagelkerke R2 and χ² into Tables 4 and 5 (original Tables 3 and 4).
Comment
In the abstract, authors stated that the different directions of association between bully involvement and facial emotion recognition must be considered when developing prevention and intervention programs. However, they did not address this point in the discussion section. How facial emotion recognition should be considered? Are there already programs working with facial emotion recognition in samples of ASD and adolescent with typical neurodevelopment? Given that the journal is focus on public health, what are the implications of the study in those terms?Response
Thank you for your suggestion. We have added a new paragraph introducing the program proposed by a previous study that enhance theory of mind performance and facial emotion recognition as below. Please refer line 292-301.
“Previous interventions against school bullying have often used universal bullying prevention programs (e.g., the Olweus Bullying Prevention Program) as the criterion standard [45]. Other recent intervention modes, including the “Learning Together” intervention program posed by Bonell et al. [46], have focused on improving social and emotional skills. However, these programs did not anchor with the core symptoms of ASD. Research found that the training program focusing on enhancing theory of mind performance ability could reduce the risk of bullying victimization in children and adolescents with high-functioning ASD [47]. Facial emotion recognition is one of the skills trained to enhance theory of mind performance ability [47].”

Reviewer 2 Report
This is an interesting study with some useful data, which does need some revision before publication.
Introduction:
line 49 prevalence rates vary greatly be definition, frequency criterion, reference period, etc … so if it is felt important to give percentages (it may not be) then more context is needed to make sense of them.
line 54 also line 61 why ‘Aggress. Behav.s’ rather than simply ‘aggressive behaviors’?
line 58 maybe ‘Several studies have …’ rather than just ‘Research …’
lines 84-86 why this hypothesis? It could be relevant to cite research suggesting some bullies are socially skills, e.g.
Sutton, J., Smith, P.K., & Swettenham, J. (1999a). Bullying and "theory of mind": a critique of the "social skills deficit" view of anti-social behaviour. Social Development, 8,117-127.
Sutton, J., Smith, P. K., & Swettenham, J. (1999b). Social cognition and bullying: Social inadequacy or skilled manipulation? British Journal of Developmental Psychology, 17, 435-450.
Peeters, M., Cillessen, A.H.N., & Scholte, R.H.J. (2010). Clueless or powerful? Identifying subtypes of bullies in adolescence. Journal of Youth and Adolescence, 39, 1041-1052.
lines 86-89 the selection of age, depression and anxiety as control variables seems arbitrary. There are many factors ‘related to bullying involvement’ (line 87) – gender, and self-esteem, for example. It is not clear why any control factors are needed, but if done, why these?
Later we are told about measures of inattention, hyperactivity, and oppositional defiance. Why? And if included they should be covered in the Introduction.
Method:
OK, but lines 152-157 see comment above
Line 158 more details of procedure should be given, e.g. were questionnaires given individually? In what order? How long did it take?
Results:
For this kind of data set, it would be really valuable to separate out ‘pure bullies’, ‘pure victims’, ‘bully/victims’, and ‘not involved’. A lot of research shows how the bully/victims are a special category. Ignoring them confounds the assessment of bullies, and victims. Only much later (lines 263-266) are we told that numbers were too small for this. Well, what were the numbers? Are they really too small? I would in any event suggest reporting emotion scores for these 4 categories. You could later move on to the analyses you have, explaining that the bully/victims are too few to analyze separately (so add them back into bullies or victims, or, if they are really few, exclude them, so you are looking at ‘pure’ bullies and victims).
195-202 what are all these models in the Tables? This was not properly explained.
Discussion:
The findings are interesting and well discussed. But the abilities of bully/victims is a major issue that must be faced directly.
Author Response
Introduction
Comment
line 49 prevalence rates vary greatly be definition, frequency criterion, reference period, etc … so if it is felt important to give percentages (it may not be) then more context is needed to make sense of them.
Response
Thank you for your suggestion. We have revised this sentence as below and delete the rates of bullying involvement. Please refer to line 49-51.
“A review article by Maïano et al. [11] stated that school‐aged youth with ASD are at greater risks of school victimization and verbal bullying than their peers without ASD.”
Comment
line 54 also line 61 why ‘Aggress. Behav.s’ rather than simply ‘aggressive behaviors’?
Response
Thank you for your suggestion. We have revised “Aggress. Behav.s” into “aggressive behaviors.” Please refer to line 55-56 and line 63-64.
Comment
line 58 maybe ‘Several studies have …’ rather than just ‘Research …’
Response
We have revised it. Please refer to line 58.
Comment
lines 84-86 why this hypothesis? It could be relevant to cite research suggesting some bullies are socially skills, e.g.:
Sutton, J., Smith, P.K., & Swettenham, J. (1999a). Bullying and "theory of mind": a critique of the "social skills deficit" view of anti-social behaviour. Social Development, 8,117-127.
Sutton, J., Smith, P. K., & Swettenham, J. (1999b). Social cognition and bullying: Social inadequacy or skilled manipulation? British Journal of Developmental Psychology, 17, 435-450.
Peeters, M., Cillessen, A.H.N., & Scholte, R.H.J. (2010). Clueless or powerful? Identifying subtypes of bullies in adolescence. Journal of Youth and Adolescence, 39, 1041-1052.
Response
Thank you for your suggestion. We have revised the contents of Introduction section to strengthen the hypotheses. We also citated the results of studies that the reviewers suggested in the new contents. Please refer to line 85-97.
“Previous studies regarding teenagers’ bullying have indicated that bullying roles and behaviors are diverse. Different bullying behaviors involve teenagers’ skills, status, and social behaviors [23]. Sutton et al. [24] claimed that a portion of the bullying perpetrators are of high social intelligence and have superior theory of mind, because lack of social intelligence often results in ineffective bullying behaviors [24]. Peeters et al. [23] also identified that a portion of the bullying perpetrators had higher social cognition than the victims did. Therefore, the perpetrators and victims may exhibit different social cognition levels [23]. This study divided bullying roles into perpetrators and victims for further exploration. We hypothesized that adolescents with high-functioning ASD who were bullying perpetrators had better facial emotion recognition than those who were non-perpetrators, whereas high-functioning ASD who were bullying victims had worse facial emotion recognition than those who were non-victims.”
Comment
lines 86-89 the selection of age, depression and anxiety as control variables seems arbitrary. There are many factors ‘related to bullying involvement’ (line 87) – gender, and self-esteem, for example. It is not clear why any control factors are needed, but if done, why these? Later we are told about measures of inattention, hyperactivity, and oppositional defiance. Why? And if included they should be covered in the Introduction.
Response
Thank you for your suggestion. We have added the reasons why we considered sex, age, and severity of ADHD, depression, and anxiety as covariates in the present study as below. Please refer to line 98-105.
“ASD is often comorbid with attention deficit hyperactivity disorder (ADHD), depression, and anxiety. Previous studies have discovered that ADHD [2, 25], depression [2, 26], and anxiety [2, 27] may be risk factors of bullying involvement and facial emotion misrecognition. In addition, sociodemographic data, such as sex [28] and age [29, 30], are often considered relevant to bullying involvement. Therefore, this study controlled for sex, age, and severity of ADHD, depression, and anxiety to identify the correlation between facial emotion recognition and bullying involvement.”
Method:
Comment
lines 152-157 see comment above
Response
Thank you for your suggestion. We have added the reasons why we considered severity of ADHD symptoms as covariates as above. Please refer to line 98-105.
Comment
Line 158 more details of procedure should be given, e.g. were questionnaires given individually? In what order? How long did it take?
Response
Thank you for your suggestion. We have added introduction for the procedures of study in the methods section as below. Please refer to line 196-202.
“Participants with ASD referred by outpatient clinics were individually provided with explanation of the purpose, procedure, and privacy policy of this study by the research assistant before the participants signed a consent form. The participants’ parents were asked to respond to the questionnaire for sociodemographic data and SNAP-IV; adolescents were asked to respond to the C-SBEQ, C-CES-D, MASC-T, and C-FERT. The overall response process required approximately 35–45 minutes.”
Results
Comment
For this kind of data set, it would be really valuable to separate out ‘pure bullies’, ‘pure victims’, ‘bully/victims’, and ‘not involved’. A lot of research shows how the bully/victims are a special category. Ignoring them confounds the assessment of bullies, and victims. Only much later (lines 263-266) are we told that numbers were too small for this. Well, what were the numbers? Are they really too small? I would in any event suggest reporting emotion scores for these 4 categories. You could later move on to the analyses you have, explaining that the bully/victims are too few to analyze separately (so add them back into bullies or victims, or, if they are really few, exclude them, so you are looking at ‘pure’ bullies and victims).
Response
Thank you for your suggestion. In the revised manuscript we have added a new paragraph to describing the procedures of grouping the participants and the results of analysis using four groups of pure perpetrators, pure victims, perpetrators/victims, and neutrals as below in the methods section and Table 1. Please refer to line 204-213.
“In the initial exploration of the data we first divided the participants into four groups, namely pure perpetrators, pure victims, perpetrators/victims, and neutrals (Table 1). The results revealed that the differences in C-FERT among the four groups did not reach the significant level. It was possibly caused by the small numbers of participants in the groups of perpetrators/victims and pure victims. Therefore, we regrouped the participants and analyzed facial emotion recognition between bullying perpetrators and non-perpetrators as well as between bullying victims and non-victims. Participants who answered 2 or 3 on any item among items 1 to 8 and items 9 to 16 from C-SBEQ were identified as bullying victims and perpetrators, respectively.”
Comment
195-202 what are all these models in the Tables? This was not properly explained.
Response
Thank you for your suggestion. We have explained all these models in the Table 4 and Table 5 as below. Please refer to line 249-258.
" In Model I to Model V, the correct rate and reaction time of C-FERT differentiation, correct rate of C-FERT ranking, as well as correct rate and reaction time of C-FERT rating were selected to multiple logistic regression as independent variables, respectively. Age, gender, depression, anxiety, inattention, hyperactivity/impulsivity and opposition were selected as control variables. The results of Models IV and V in Table 4 indicated that bullying perpetrators performed significantly better on rating intensity of facial emotion in the C-FERT rating process than nonperpetrators did. The results of Model III in Table 5 indicated that bullying victims performed significantly worse on ranking intensity of facial emotion in the C-FERT ranking process than nonvictims."
Discussion:
Comment
The findings are interesting and well discussed. But the abilities of bully/victims is a major issue that must be faced directly.
Response
Thank you for your suggestion. We agreed that facial emotion recognition in perpetrators/victims warranted further study. We have listed the small sample size for perpetrators/victims as one of limitations of this study as below. Please page 15, line 327-333.
“Moreover, the small number of participants who were both bullying perpetrators and victims (perpetrator-victims) limited the possibility of this study to examine whether the perpetrator-victims possess unique patterns of facial emotion recognition. Further study should be conducted to realize the role of facial emotion recognition ability in relation to bullying involvement among adolescents with high-functioning ASD when their bullying involvements are divided into four categories.”

Round 2
Reviewer 1 Report
Authors have addressed all my concerns regarding the first version of the manuscript. Hypothesis are now better explained and based on previous research. Methos is well-described allowing replicability. Neccesary statistics are now included. Conclusions are drawn from the results. In my opinion, this study merits publication.
Author Response
Thank you for your comment.
Reviewer 2 Report
I think the authors have responded well to the earlier review.
I would just suggest one change, at lines 53-55. Whether bullies are more depressed etc is somewhat controversial. Findings such as those cited are more likely when 'pure' bullies and bully/victims are put together (as is done in this study ...). When pure bullies are considered, the findings are much less clear, and some researchers, such as Volk, and Wolke, argue that bullies are doing well and that bullying can be an adaptive strategy.
So I suggest either expand these few lines to put both points of view, or simply say 'Some researchers argue that bullying perpetrators ...'
Author Response
Thank you so much for providing us an opportunity to revise our manuscript. We also appreciated reviewers’ comments. As discussed below, we have revised our manuscript with underlines according to reviewers’ suggestions. Please let me know if we need to provide anything else regarding this revision.
Response to comments
Comment
I would just suggest one change, at lines 53-55. Whether bullies are more depressed etc is somewhat controversial. Findings such as those cited are more likely when 'pure' bullies and bully/victims are put together (as is done in this study ...). When pure bullies are considered, the findings are much less clear, and some researchers, such as Volk, and Wolke, argue that bullies are doing well and that bullying can be an adaptive strategy.
So I suggest either expand these few lines to put both points of view, or simply say 'Some researchers argue that bullying perpetrators ...'
Response
Thank you for your suggestion. We have revised this sentence as below. Please refer to line 53-59.
“Although Volk et al. proposed that bullying perpetration may be the result of an evolutionarily adaption for somatic resources, mates, and dominance [7], some researchers argued that both pure bullying perpetrators and perpetrator-victims have higher risks to develop mental health problems [2, 8]. A previous study also reported that both pure perpetrators and perpetrator-victims were more likely to report suicidal ideation and attempt than the neutral group [9]. ”